# Cohort profile: The UK COVID-19 Public Experiences (COPE) prospective longitudinal mixed-methods study of health and well-being during the SARSCoV2 coronavirus pandemic

Rhiannon Phillips[1]*, Khadijeh Taiyari[2], Anna Torrens-Burton[3], Rebecca Cannings-John[2], Denitza Williams[1], Sarah Peddle[4], Susan Campbell[4], Kathryn Hughes[3], David Gillespie[2], Paul Sellars[1], Bethan Pell[5], Pauline Ashfield-Watt[6], Ashley Akbari[7], Catherine Heidi Seage[1], Nick Perham[1], Natalie Joseph-Williams[3], Emily Harrop[8,9], James Blaxland[1], Fiona Wood[3], Wouter Poortinga[10,11], Karin Wahl-Jorgensen[9], Delyth H. James[1], Diane Crone[1], Emma Thomas-Jones[2], Britt Hallingberg[1]

1 Cardiff School of Sport and Health Sciences, Cardiff Metropolitan University, Cardiff, United Kingdom, 2 Centre for Trials Research, Cardiff University, Cardiff, United Kingdom, 3 Division of Population Medicine, PRIME Centre Wales, Cardiff University, Cardiff, United Kingdom, 4 Public and Patient Partner, Cardiff, United Kingdom, 5 Centre for the Development and Evaluation of Complex Intervention for Public Health Improvement (DECIPHer), Cardiff University, Cardiff, United Kingdom, 6 Division of Population Medicine, HealthWise Wales, Cardiff University, Cardiff, United Kingdom, 7 Population Data Science, Health Data Research UK, Swansea University Medical School, Swansea University, Swansea, United Kingdom, 8 Division of Population Medicine, Marie Curie Palliative Care Research Centre, Cardiff, United Kingdom, 9 Cardiff School of Journalism, Media and Culture, Cardiff University, Cardiff, United Kingdom, 10 Welsh School of Architecture, Cardiff University, Cardiff, United Kingdom, 11 School of Psychology, Cardiff University, Cardiff, United Kingdom

* RPhillips2@cardiffmet.ac.uk

**Data Availability Statement:** Data cannot be shared publicly because of the data is of a detailed

## Abstract

Public perceptions of pandemic viral threats and government policies can influence adherence to containment, delay, and mitigation policies such as physical distancing, hygienic practices, use of physical barriers, uptake of testing, contact tracing, and vaccination programs. The UK COVID-19 Public Experiences (COPE) study aims to identify determinants of health behaviour using the Capability, Opportunity, Motivation (COM-B) model using a longitudinal mixed-methods approach. Here, we provide a detailed description of the demographic and self-reported health characteristics of the COPE cohort at baseline assessment, an overview of data collected, and plans for follow-up of the cohort. The COPE baseline survey was completed by 11,113 UK adult residents (18+ years of age). Baseline data collection started on the 13th of March 2020 (10-days before the introduction of the first national COVID-19 lockdown in the UK) and finished on the 13th of April 2020. Participants were recruited via the HealthWise Wales (HWW) research registry and through social media snowballing and advertising (Facebook®, Twitter®, Instagram®). Participants were predominantly female (69%), over 50 years of age (68%), identified as white (98%), and were living with their partner (68%). A large proportion (67%) had a college/university level education, and half reported a pre-existing health condition (50%). Initial follow-up plans for

and sensitive nature, and public contributors and research participants expressed concerns about privacy and security during the development and recruitment stages of this research. Data are available from the Cardiff Metropolitan University Applied Psychology Ethics Committee (contact via HealthEthics@cardiffmet.ac.uk) for researchers who meet the criteria for access to confidential data.

**Funding:** The initial stages of this research (March 2020 – August 2020) were supported by internal resources at Cardiff Metropolitan University (www. cardiffmet.ac.uk), Cardiff University (www.cardiff. ac.uk), HealthWise Wales (https://www. healthwisewales.gov.wales), and PRIME Centre Wales (http://www.primecentre.wales). This included allowing core team members time to design, set up, and conduct the baseline and 3-month data collection. Financial support was provided by internal Cardiff Metropolitan University 'Get Started' and Cardiff University Division of Population funds to support transcription of the baseline qualitative data. PRIME Centre Wales, HealthWise Wales and the Centre for Trials Research are part of Health and Care Research Wales infrastructure (https:// healthandcareresearchwales.org). Health and Care Research Wales is a networked organisation supported by Welsh Government. In August 2020, a Sêr Cymru III Tackling COVID-19 grant (https:// gov.wales/ser-cymru, Project number WG 90) was awarded to cover our follow-up data collection, analysis and dissemination activities for the period between the 1st of August 2020 to 31st of March 2021. This work is supported by Health Data Research UK, which receives its funding from HDR UK Ltd (HDR-9006) funded by the UK Medical Research Council, Engineering and Physical Sciences Research Council, Economic and Social Research Council, Department of Health and Social Care (England), Chief Scientist Office of the Scottish Government Health and Social Care Directorates, Health and Social Care Research and Development Division (Welsh Government), Public Health Agency (Northern Ireland), British Heart Foundation (BHF) and the Wellcome Trust. The funders had no role in study design, data collection and analysis, decision to publish, or preparation of the manuscript.

**Competing interests:** The authors have declared that no competing interests exist.

the cohort included in-depth surveys at 3-months and 12-months after the first UK national lockdown to assess short and medium-term effects of the pandemic on health behaviour and subjective health and well-being. Additional consent will be sought from participants at follow-up for data linkage and surveys at 18 and 24-months after the initial UK national lockdown. A large non-random sample was recruited to the COPE cohort during the early stages of the COVID-19 pandemic, which will enable longitudinal analysis of the determinants of health behaviour and changes in subjective health and well-being over the course of the pandemic.

## Introduction

The COVID-19 pandemic is having a profound and wide-reaching effect on societies globally [1]. Public perceptions of pandemic viral threats and government policies can influence adherence to containment, delay, and mitigation policies such as physical distancing, hygienic practices, use of physical barriers (such as face coverings, face shields, protective clothing, and disposable gloves), and uptake of testing, contact tracing, and vaccination programs [2–14]. There are marked social inequalities in the risk of harm to health and well-being during the COVID-19 pandemic, particularly in relation to ethnicity, occupational status, social deprivation, sex, housing, and pre-existing physical and mental-health conditions [15–18]. Understanding the impact of the pandemic and related policies on physical health and psychological well-being is a high priority for government and public health agencies [1, 19]. Timely, high-quality research that adopts a holistic approach to behaviour, health, and well-being is needed to inform the immediate response to and long-term recovery from the COVID-19 pandemic [1, 18].

The COVID-19 UK Public Experiences (COPE) study is a prospective longitudinal mixed-methods study that was established during the early stages of the pandemic outbreak, which aimed to build a detailed understanding of health behaviour and health and well-being outcomes over the course of the pandemic [20]. The Capability, Opportunity, Motivation model of behaviour (COM-B) model was selected as the conceptual model for the COPE study to provide a systematic method for identifying potentially modifiable determinants of health behaviour. The COM-B forms a core part of the Behaviour Change Wheel (BCW) framework for the development and evaluation of complex behaviour change intervention, enabling mapping of determinants of behaviour to classes of intervention and specific behaviour change techniques [21]. The COM-B is an integrated model that takes into account multiple factors that can influence behaviour, such as knowledge, beliefs, attitudes, and practical barriers and facilitators in understanding health behaviour [21]. The COM-B has been used to explain a range of infection-related health behaviour, including hand hygiene, environmental disinfection, use of personal protective equipment (PPE), uptake of screening and testing, use of antivirals and antibiotics for respiratory tract infections, uptake of influenza vaccines, and lifestyle behaviour in the context of respiratory tract infection outbreaks [10, 13, 22–44].

Infection-transmission prevention behaviours, including hygiene behaviours, social distancing, use of physical barriers, and uptake of vaccinations (when they became available) were of primary interest in the COPE study. The COM-B provided a useful framework for understanding the complex relationship between multiple determinants of infection-transmission prevention behaviour and the wider social, political, and environmental context [11]. Based on research in previous pandemics [45–47], it was anticipated that psychological

capability (e.g., knowledge and skills), opportunity (e.g., physical and social environment), and motivation (e.g., attitudes, appraisal of risk, fear) were potentially important determinants of these behaviours. The COM-B has also been applied to understanding changes in health behaviour more generally, such as physical activity, in the context of a pandemic [41]. Key health behaviours, including physical activity, healthy eating, smoking, alcohol use, socializing, relaxing activities, and health service use were also of interest in the COPE study due to the likely disruption of these behaviours in the context of the pandemic and the potential impact on health and well-being outcomes [20]. In terms of identifying determinants of these behaviours over the course of the pandemic, we were particularly interested in the effects of social and physical environment (opportunity) and whether the perceived seriousness of the COVID-19 threat increased or decreased engagement with health promoting behaviour (motivation).

This paper describes the initial profile of the COPE study online cohort, providing an overview of the collected data, a description of the initial demographic and self-reported health characteristics of the population reported during the baseline assessment, and detailing plans for follow-up of the cohort.

## Materials and methods

### Design

The COPE study is a longitudinal mixed-methods prospective cohort study [20]. The COM-B model was used to identify potential determinants of infection-transmission prevention behaviour and key health behaviours during the COVID-19 pandemic. This informed the selection of measures at each data collection point, provided a framework for planning analysis and data triangulation, facilitated interpretation of findings, and provided a systematic method for identifying potential opportunities for interventions.

### Setting

The COPE study focused on understanding health behaviour over the course of the COVID-19 pandemic in a UK community setting. The World Health Organization (WHO) declared the COVID-19 outbreak a pandemic on the 11[th] of March 2020 [48]. On the 13[th] of March 2020 when the COPE baseline survey was launched, within the UK there had been 480 confirmed cases and 16 reports of people having died within 28 days of having had a positive COVID-19 test (Fig 1). When the COPE baseline survey closed on the 13[th] of April 2020, there had been 4,168 confirmed COVID-19 cases and 895 deaths within 28 days of a positive COVID-19 test in the UK [49]. Globally, there had been 533,132 confirmed cases of COVID-19 and 51,585 deaths by the time the COPE baseline survey closed [50].

### Cohort recruitment

The COPE cohort was recruited through a baseline online survey. Recruitment occurred via two routes:

1. A multi-faceted sampling method based on convenience sampling, snowballing, and purposive sampling via social media. We created dedicated Facebook[®] (@COVID19publicexperiencesUK), Instagram[®] (@COVID19publics1) and Twitter[®] (@COVID19publics1) feeds, and a study website (https://copestudy.yolasite.com) to publicise the study. Through these, we regularly posted information about the study and invitations to take part in the baseline survey, which included a hyperlink to the online survey. Social media feeds were regularly monitored and moderated. Facebook and Instagram's paid promotion feature

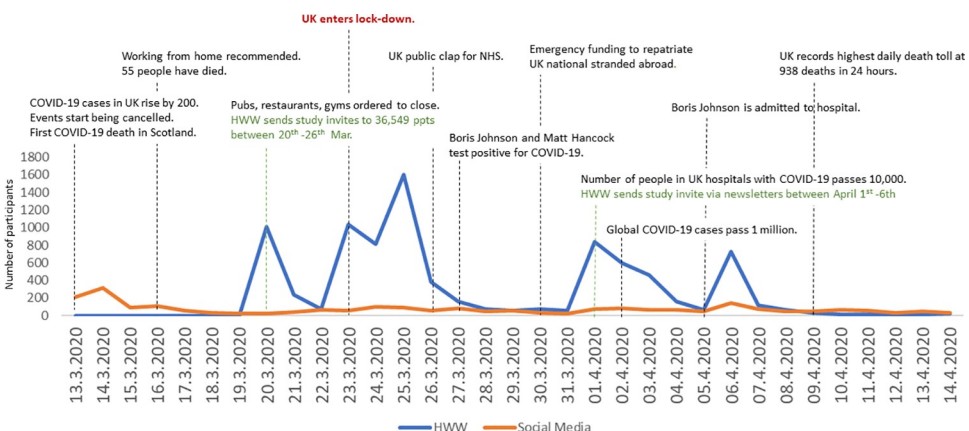

**Fig 1. Number of participants completing the COPE survey via the HealthWise Wales (HWW) and social media (SM) routes on each day of the baseline recruitment period.**

was used to boost posts. Based on research in previous pandemics, we anticipated that key demographic variables, such as age, gender, and education, were likely to be associated with the impact that the pandemic had on individuals and on their responses to the pandemic [3, 51, 52]. It became clear during the early stages of recruitment that men and younger age groups were under-represented in our cohort, so we tailored as recruitment progressed to focus on using our finite resources to increase our reach with these demographic groups (total advertising budget for the study, £150).

2. From the 20th of March onwards, the study was advertised via HealthWise Wales (HWW) [53], a national population survey and research register of participants who live or receive healthcare in Wales. Invitations to take part were e-mailed to HWW participants on two occasions, providing a summary of the COPE study and a hyperlink to the survey.

## Follow-up procedures for the longitudinal cohort study

**Longitudinal survey data collection points.** There was much uncertainty about the course of the pandemic when the study was launched. A balance needed to be found between the need to capture detailed, comprehensive data without over-burdening participants. Therefore, survey data collection at three static time points were planned for the COPE Study in the first instance. The baseline survey took place between the 13th of March and the 13th of April 2020, as the first wave of COVID-19 was occurring in the UK and lockdown measures were coming into force. Follow-up surveys were planned at three months (June/July 2020) and 12 months (March/April 2021) after the initial lockdown to provide data on short- and medium-term changes in behaviour and health and well-being outcomes. Our three-month survey data collection point coincided with the initial easing of lockdown restriction after the peak of the first wave in the UK had subsided. The 12-month data collection point occurred following a second wave and prolonged lockdown period in the UK, while restrictions were slowly being eased and a mass vaccination program was underway. As the pandemic progressed, it became clear that there would be a significant longer-term impact on health and well-being of the general population, and that the recovery phase had not been reached by the 12-month follow-up survey. Therefore, data linkage with electronic health record (EHR) data sources and HWW data, and further survey data collection points at 18 and 24-months after the initial UK

lockdown were added to the study protocol to enable longer-term behavioural and health and well-being outcomes to be assessed.

**Data collection procedure.** Baseline data were collected using the Onlinesurveys.ac.uk platform. Participants who consented to follow-up provided their e-mail address and/or a telephone number. Follow-up data will be collected using Qualtrics.com for pragmatic reasons. Participants were sent an individualised link to their e-mail or mobile phone which recorded a unique study identification number alongside responses to enable data linkage from each time point to take place. At each follow-up point, participants were sent an initial invitation and non-responders received a maximum of two reminders.

## Measures

The COM-B model was used to inform the selection of measures for this study, with regular review of measures as the pandemic progressed based on information gathered via the qualitative component of the COPE study, feedback from the study patient and public involvement partners, and the shifting pandemic, policy, and social context. A summary of topics covered at each time point is provided in Table 1. S2 File provides details of all items included in the surveys along with citations for the original source of items/scales where applicable. Further information on the development of the surveys and specific measures included at each time point is provided in the study protocol [20]. In this article, we focus on characterising the COPE cohort at the baseline assessment in terms of key demographic characteristics and self-reported health.

**Demographic data.** Data were collected at baseline on age category, gender, highest level of education, ethnicity, marital status, and caring responsibilities (children under 18-years of age, children under the age of 5-years, children with pre-existing health conditions, adults with pre-existing health conditions, older adults).

**Self-reported health and well-being.** To assess perceived exposure to COVID-19, participants were asked whether they had or thought they may have had COVID-19, and whether they had experienced the key symptoms of COVID-19 that had been identified in the early stages of the pandemic (continuous cough, fever, loss of taste or smell). Participants were asked whether they had pre-existing medical conditions, which conditions these were, and whether they had received a seasonal flu vaccination in the last 12-months.

Four items were included from the SF12v1 measure [54] to assess general health and mental health. The general health item was scored from 1 = poor to 6 = excellent. Subjective well-being was measured using three items to assess vitality; "Did you have a lot of energy?", and mental health; "Have you felt calm and peaceful?", "Have you felt downhearted and blue" (reversed). The summed score of the two mental health items has been found to be a useful screening tool in the general population for affective disorders [55].

**Data analyses.** Recruitment of participants through HWW and social media on each day that the baseline survey was open was mapped against key events relating to COVID-19 in the UK to provide the context for data collected within a rapidly changing situation. Descriptive analysis of the baseline demographic characteristics, self-reported health and well-being was carried out for participants overall, and for those joining the COPE cohort from each of the recruitment routes (HWW and social media) to provide a detailed profile of the cohort. Population data for Wales and the UK was gathered from published sources to enable us to understand which demographic groups may be over- or under-represented in the COPE cohort. Analyses were conducted in SPSSv27.

## Patient and public involvement (PPI)

Members of the public were consulted informally to comment on the online baseline survey design, which needed to be developed rapidly before we had secured any external funding for

**Table 1. Summary of topics covered at each time point of the COPE longitudinal survey.**

| Topic | Baseline: Mar/Apr 2020 | 3-month follow-up: Jun/Jul 2020 | 12-month follow-up: Mar/Apr 2021 |
|---|---|---|---|
| **Health Behaviour** | | | |
| COVID-19 transmission-prevention behaviour (including physical distancing, hygiene, and use of physical barriers) | X | X | X |
| Health behaviour (including smoking, diet, alcohol, physical activity, social contact, relaxing activities) | X | X | X |
| COVID-19 vaccination uptake | | | X |
| COVID-19 testing and self-isolation | | | X |
| **Capability** | | | |
| COVID-19 knowledge and information needs | X | | |
| Sources of information on COVID-19 accessed and perceived reliability of these sources | X | X | X |
| Perceptions of barriers and facilitators to engaging in infection-transmission prevention behaviour | X | X | X |
| **Motivation** | | | |
| COVID-19 risk perception (perceived susceptibility, harmfulness, worry and attention) | X | X | X |
| Self-efficacy for reducing COVID-19 transmission | | X | X |
| Perceived barriers and facilitators to engaging in infection-transmission prevention behaviour | X | X | X |
| Attitudes towards COVID-19 vaccination | | X | X |
| Attitudes towards COVID-19 community testing and contact tracing | | X | |
| Attitudes towards self-isolation | | | X |
| **Opportunity (social and physical environment)** | | | |
| Sex, age, pre-existing medical conditions, education, religion, ethnic group, sexual orientation | X | | |
| Employment status | X | X | X |
| Caring responsibilities | X | | X |
| Neighbourhood cohesion | | X[a] | X |
| Perceived access to green spaces | | X[a] | |
| Work environment | X | X | X |
| Bereavement | | X[a] | |
| Patient-reported healthcare experiences (including patient-reported safety concerns) | | X[a] | X |
| **Health and well-being outcomes** | | | |
| COVID-19 incidence & symptoms | X | X | X |
| Subjective physical and mental health: Four items from the 12-item version of the Short-Form Health Survey (SF12v1) [54] at all time points. | X | X | X |
| Patient Health Questionnaire (PHQ-4) added at 3 and 12-month follow-up. | | | |

[a]Included as optional modules in the 3-month follow-up survey.

the project. We subsequently invited expressions of interest in joining the study team from the Centre for Trials Research and PRIME Centre PPI panels and two members of the public formally joined our research team. They commented on the design of the study, including the survey and interview questions in each phase, were co-applicants on the funding application, and are included as co-authors on key study outputs (including for this manuscript), and will support wider dissemination of findings as the research progresses.

## Ethics and governance

Ethical approval was obtained for the COPE Study from the Cardiff Metropolitan University Applied Psychology ethics panel on 13.3.20 (Project reference Sta-2707). Participants provided

consent and confirmed eligibility electronically at the beginning of the baseline survey and provided consent to re-contact at the end of the survey. Additional consent was sought from participants recruited via HWW at 3 and 12-month follow up for linkage their COPE data with data held by HWW and their electronic healthcare records (EHRs) at 3 and 12-month follow up (Project Sta-2707, amendment 2, approved 11.6.2020). Participants were asked for consent to recontact for longer-term follow-up during the 12-month survey (Project Sta-2707, amendment 3, approved 5.3.2021). The HWW research database received favourable ethical reviews 15_WA_0076 and 20_WA_0064. An application from COPE Cymru to access HWW data on mutual participants (with participant consent) within the Secure Protected Portal and HWW Resource SAPPHIRe was approved on 06/02/21. Approvals for provision of healthcare data to HWW relevant to the COPE were granted by an independent Information Governance Review Panel (IGRP) under project 0415 HWW on 15/12/20 and 02/03/21.

## Findings to date

**Baseline cohort characteristics.**    11,113 people took part in the baseline COPE survey between March 13th and April 12th 2020. Recruitment for the baseline survey took place during a time of rapid change in infection rates and government policy, as illustrated in Fig 1.

The majority of the COPE cohort were recruited through HWW (n = 8,726, 79%) and as such were resident or receiving healthcare in Wales. An additional n = 2,386 (21%) recruited via social media (UK-wide). A summary of baseline characteristics of the COPE cohort is provided in Table 2. Table 2 also provides a summary of Welsh and UK population data available from published sources to enable comparison of the COPE cohort with the general population and identification of under-represented demographic groups within our cohort.

Females (69%), people over 60 years of age (68%), those who were married or in a civil partnership, (58%) and those who had achieved a higher education qualification (67%) were over-represented in the sample. With the majority of our sample being in Wales, 98% reported being white, which was similar to the Welsh population (96% white) [56]. Half of the COPE cohort reported a pre-existing health condition (51%), which was slightly higher than in the Welsh population (48%) where there are higher rates of people living with longstanding conditions than in England (43%). Despite the high rate of pre-existing conditions, self-evaluation of general health was better in the COPE cohort (81% rating health as good, very good, or excellent) compared with the Welsh (71%) and English (75%) populations. At least one caring responsibility for children, adults with a pre-existing health condition and/or older adults (aged 70+) was reported by 42% of the COPE cohort, and 19% had children aged <18 years living in the household. Flu vaccination for adults were routinely offered via the NHS in 2019/2020 to all those aged 65+, those aged <65 who are in clinical at-risk groups, and frontline health and social care workers. Self-reported flu vaccine uptake in the COPE cohort was lower than national averages for England and Wales for the 2019–2020 winter season, particularly in the under 60 age group who reported at least one pre-existing medical condition [57, 58].

There were some differences between participants joining the cohort through the HWW and social media routes (see S1 Table). There was increased representation from younger age groups, ethnic minority groups and those reporting caring responsibilities for children under the age of 18 years in the social media sample compared to the HWW sample. Females represented an even larger majority (82%) in the social media sample.

## Exposure to COVID-19 infection at baseline

In the COPE baseline survey, 347 (3.1%) people thought that they currently or had already had COVID-19, and 1,799 (16.2%) thought they might have had it. Only 34 (0.3%) people had

**Table 2. COPE cohort baseline characteristics and population data for published sources for Wales and the UK.**

| Characteristic | Population data sources | Category | COPE cohort (n = 11,113) | Wales (Population aged 16+ = 2,589,044) | UK (Population aged 16+ = 54,098,971) |
|---|---|---|---|---|---|
| | | | % | % | % |
| Sex | Stats Wales [56] | Male | 30 | 49 | 49 |
| | | Female | 69 | 51 | 51 |
| | | Other | <1 | Data not available | Data not available |
| Age group | StatsWales [56] | Older adults*: | 47 | 26 | 23 |
| | | Over 60 years—Welsh and UK population data age groups | | | |
| | | 65 years and over–COPE cohort age groups | | | |
| Ethnicity | StatsWales [56], Office for National Statistics [59] | White | 98 | 96 | 86 |
| | | Other | 2 | 4 | 14 |
| Marital status | Office for National Statistics [60] | Married or civil partnered | 58 | N/A | 50 |
| Highest level of education | Office for National Statistics [61] | People who have achieved a higher education qualification | 67 | 47 | 40 |
| Flu vaccination in the last 12 months* | Public Health England [57], Public Health Wales [58] | Adults <60 years with a pre-existing condition who had received a flu vaccination | 25 | 44 | 44** |
| | | Older adults who had received a flu vaccination (COPE aged 60+, UK and Welsh population aged 65+) | 62 | 69 | 72** |
| Pre-existing medical conditions | StatsWales [62], Office for National Statistics [63] | Proportion reporting any longstanding health condition(s) | 51 | 48 | 43 |
| General health: self-reported | StatsWales [62], NHS Digital [64] | COPE—good, very good, or excellent | 81 | 71 | 75** |
| | | Population data–good or very good | | | |
| | | COPE–poor | 5 | 9 | 7** |
| | | Population data—bad or very bad | | | |

*COPE age bands and general health self-evaluation categories do not correspond directly to publicly available population data and closest approximation of categories has been provided.

**Data for England rather than UK.

been diagnosed with COVID-19 by a health professional and 13 people (0.2%) had taken a laboratory test for COVID-19 with 6 (0.1%) testing positive for COVID-19. By April 13[th] 2020 there had been 96,877 confirmed cases of COVID-19 in the UK, representing around 0.15% of the UK population and the report of positive COVID-19 tests within our cohort was consistent with this. The COPE baseline data were collected before widespread community testing for COVID-19 was available in the UK and it is likely that there were more unconfirmed and asymptomatic cases present during this period in our cohort and the general population than those confirmed by laboratory testing.

## Consent to follow-up

Overall, 9,899 (89.1%) of the 11,113 COPE participants consented to follow-up surveys at the end of their baseline survey. Rates of consent to follow-up were higher amongst those recruited via HWW (8,126/8,727, 93.1%) than via social media (1,773/2,386, 74.3%). Consent to contact for qualitative interviews was provided by 4,833 participants (43.5%). Additional explicit consent for data linkage with HWW and EHR data will be sought from HWW participants during follow-up surveys, opening the potential for analysis of pre-pandemic health, well-being, and behavioural data and long-term follow-up of health outcomes and health service usage in this cohort.

## Discussion

The COPE cohort was established during the early stages of the COVID-19 outbreak in the UK before the initiation of lockdown and the likely impact of the pandemic was known. Over 11,000 people were recruited into the cohort during the baseline survey, with 89.1% consenting to follow-up. The COPE study was designed by a multi-disciplinary team using an established theoretical framework, the COM-B model, to guide the measures used, analysis and interpretation of findings. Our analysis at each survey time point will be considered within the context of local, national and international policy and infection rates [49, 50, 65, 66], and major themes identified in mainstream media at the time of data collection. This will enable us to build a detailed understanding of health behaviour and subjective health and well-being in this cohort as the pandemic progresses.

### Future plans

Analysis of the cross-sectional and longitudinal data collected from this cohort will be conducted to identify the characteristics of people who have low engagement with infection-transmission prevention behaviour at different stages in the pandemic, with an emphasis on understanding the role of COVID-19 risk perception in determining behaviour. We will examine broader changes in lifestyle, health and well-being to identify groups that may be in need of additional support during lockdown periods, and to inform the design and implementation of interventions to promote health, well-being and re-engagement with social roles during the recovery phase.

Tailored behavioural measures needed to be developed and adapted as the pandemic progressed to capture data on relevant and emerging issues, such as community testing, contact tracing, vaccination, and patient safety. Data from the 3-month follow-up survey (not presented here) will be analysed to investigate the factor structure and conduct validation of the behavioural measures used in this study. Establishing what is 'appropriate' behaviour at different stages of the pandemic will also be considered during analysis and interpretation of findings, as people will need to adjust their behaviour as the seriousness of the COVID-19 threat shifts and lockdown restrictions tighten and relax over the course of the pandemic. As well as our pre-planned analyses, we will draw on evidence from the rapidly growing body of research on behaviour, risk perception, and health behaviour and well-being during the COVID-19 pandemic to inform additional analyses of our data and to enable us to interpret our findings in the context of the wider literature [e.g. 1, 8, 11, 12, 41, 67–81].

Public and patient involvement in research is essential in ensuring that research is relevant to participants, designed in a way that is acceptable to people taking part, and improves the communication of results to participants as they become available [82]. Public involvement in this study has been essential to designing questionnaires and interview schedules that are relevant and acceptable to participants, designing information sheets and consent procedures, and communicating findings of this research. We will continue to involve our patient and public partners who are joint grant-holders for this project and included as authors on all major outputs. We will continue to regularly communicate with our research participants, both gathering and acting on their feedback as follow-up data is collected and communicating the findings of the study as soon as they become available.

### Limitations

We did not employ a random sampling technique in this study due to resource and time constraints, as we needed to rapidly establish the cohort during the early stages of the pandemic to capture vital baseline data. Our study population was self-selecting and comparison with Welsh and UK population data from published sources indicated that our sample is not

representative of the Welsh or UK general population in terms of their demographic profile. Females, those of white ethnicity, and those with higher education qualifications are over-represented in the COPE cohort relative to the Welsh and UK general populations. The COPE cohort also includes a higher proportion of people who are in the older age groups and have pre-existing medical conditions that in the general population, and as such are at increased risk of severe harm from COVID-19 disease, making this an important population to study. There is variation within the COPE cohort in terms of demographic characteristics, caring responsibilities, physical and mental health, to enable us to capture a wide range of views and experiences, and to carry out meaningful and novel analysis with the data produced. However, the characteristics of the cohort need to be considered during analysis of data and interpretation, particularly with regard tour ability to generalise from our findings.

The majority of the COPE cohort were recruited via HWW and were therefore resident and/or accessing healthcare in Wales. Responsibility for healthcare is devolved to the Welsh Government, and policy has diverged from the rest of the UK in terms of the timing and implementation of COVID-19 infection-transmission prevention policy to fit the needs of the local population and healthcare services [65]. Though implemented slightly differently, the measures introduced to control the spread of COVID-19 have been broadly similar in Wales to the rest of the UK over the course of the pandemic to date, including guidance on hygiene and social distancing, stay at home or stay local guidelines, local lockdowns, travel restrictions, school closures, closure of non-essential shops and services during peak infection periods, rapid implementation of an adult vaccination program in 2021 [65]. Trust in UK Government and Welsh Government has fluctuated over the course of the pandemic, but trust in the Welsh Government has generally been higher than for the UK Government, which may be reflected in the vaccine attitudes reported in the COPE study [83, 84]. The policy context in devolved nations as well as UK-wide policy will need to be considered when interpreting our findings.

## Conclusions

The COPE study enables the utilisation of a cohort with COVID specific details, linked together in a reproducible dynamic platform approach to routine EHR data, to enable opportunities to conduct anonymised person level in-depth longitudinal analysis of health behaviour, subjective health and well-being, and their determinants in a large UK community sample over the course of the COVID-19 pandemic. The socio-demographic characteristics of this cohort and the socio-political context during data collection windows will be considered during data analysis and interpretation of findings.

## Supporting information

**S1 File. COPE baseline survey questions.**
(PDF)

**S2 File. COPE data dictionary.**
(XLSX)

**S1 Table. UK and Welsh population demographic data.**
(DOCX)

## Acknowledgments

We would like to thank the participants who have taken part in the COPE study and volunteered their time to help us understand their experiences through the pandemic. We are

grateful to our patient and public involvement members for their invaluable contributions to designing and steering this research. This study was facilitated by HealthWise Wales, the Health and Care Research Wales (HCRW) initiative which is led by Cardiff University in collaboration with the SAIL Databank, Swansea University. We are grateful for their invaluable support and expertise. This study will make use of survey and healthcare data held by HealthWise Wales. We would like to acknowledge all the data providers who make anonymised data available for research. We would like to thank Cardiff Metropolitan University, Cardiff University, PRIME Centre Wales, and Swansea University who have all been immensely supportive of this work, allowing out team the time and providing infrastructure to get the study up and running quickly during the very early stages of the pandemic.

## Author Contributions

**Conceptualization:** Rhiannon Phillips, Rebecca Cannings-John, Denitza Williams, Sarah Peddle, Susan Campbell, Kathryn Hughes, David Gillespie, Paul Sellars, Bethan Pell, Pauline Ashfield-Watt, Ashley Akbari, Catherine Heidi Seage, Nick Perham, Natalie Joseph-Williams, Emily Harrop, James Blaxland, Fiona Wood, Wouter Poortinga, Karin Wahl-Jorgensen, Delyth H. James, Diane Crone, Emma Thomas-Jones, Britt Hallingberg.

**Data curation:** Rhiannon Phillips, Anna Torrens-Burton, Pauline Ashfield-Watt, Britt Hallingberg.

**Formal analysis:** Rhiannon Phillips, Khadijeh Taiyari, Anna Torrens-Burton, Britt Hallingberg.

**Funding acquisition:** Rhiannon Phillips, Denitza Williams, Kathryn Hughes, Ashley Akbari, Catherine Heidi Seage, Nick Perham, Fiona Wood, Emma Thomas-Jones, Britt Hallingberg.

**Investigation:** Denitza Williams.

**Methodology:** Rhiannon Phillips, David Gillespie, Bethan Pell, Pauline Ashfield-Watt, Ashley Akbari, Natalie Joseph-Williams, Emily Harrop, Fiona Wood, Wouter Poortinga, Karin Wahl-Jorgensen, Emma Thomas-Jones, Britt Hallingberg.

**Project administration:** Rhiannon Phillips.

**Supervision:** Rebecca Cannings-John, David Gillespie, Pauline Ashfield-Watt, Delyth H. James, Diane Crone, Britt Hallingberg.

**Writing – original draft:** Rhiannon Phillips, Britt Hallingberg.

**Writing – review & editing:** Rhiannon Phillips, Khadijeh Taiyari, Anna Torrens-Burton, Rebecca Cannings-John, Denitza Williams, Sarah Peddle, Susan Campbell, Kathryn Hughes, David Gillespie, Paul Sellars, Bethan Pell, Pauline Ashfield-Watt, Ashley Akbari, Catherine Heidi Seage, Nick Perham, Natalie Joseph-Williams, Emily Harrop, James Blaxland, Fiona Wood, Wouter Poortinga, Karin Wahl-Jorgensen, Delyth H. James, Diane Crone, Emma Thomas-Jones, Britt Hallingberg.

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
