## [Decision Letter · Decision Letter 0]

25 Aug 2021

PONE-D-21-22443

Cohort profile: The UK COVID-19 Public Experiences (COPE) prospective longitudinal mixed-methods study of health and well-being during the SARSCoV2 coronavirus pandemic

PLOS ONE

Dear Dr. Phillips,

Thank you for submitting your manuscript to PLOS ONE. After careful consideration, we feel that it has merit but does not fully meet PLOS ONE’s publication criteria as it currently stands. Therefore, we invite you to submit a revised version of the manuscript that addresses the points raised during the review process.

We look forward to receiving your revised manuscript.

Kind regards,

Ismaeel Yunusa, PharmD, PhD

Academic Editor

PLOS ONE

“The initial stages of this research (March 2020 – August 2020) were supported by internal resources at Cardiff Metropolitan University, Cardiff University, HealthWise Wales, and PRIME Centre Wales. This included allowing core team members time to design, set up, and conduct the baseline and 3-month data collection. Financial support was provided by internal Cardiff Metropolitan University ‘Get Started’ and Cardiff University Division of Population funds to support transcription of the baseline qualitative data. PRIME Centre Wales, HealthWise Wales and the Centre for Trials Research are part of Health and Care Research Wales infrastructure. Health and Care Research Wales is a networked organisation supported by Welsh Government.

In August 2020, a Sêr Cymru III Tackling COVID-19 grant (Project number WG 90) was awarded to cover our follow-up data collection, analysis and dissemination activities for the period between the 1st of August 2020 to 31st of March 2021. This work is supported by Health Data Research UK, which receives its funding from HDR UK Ltd (HDR-9006) funded by the UK Medical Research Council, Engineering and Physical Sciences Research Council, Economic and Social Research Council, Department of Health and Social Care (England), Chief Scientist Office of the Scottish Government Health and Social Care Directorates, Health and Social Care Research and Development Division (Welsh Government), Public Health Agency (Northern Ireland), British Heart Foundation (BHF) and the Wellcome Trust.”

 “The initial stages of this research (March 2020 – August 2020) were supported by internal resources at Cardiff Metropolitan University (www.cardiffmet.ac.uk), Cardiff University (www.cardiff.ac.uk), HealthWise Wales (https://www.healthwisewales.gov.wales), and PRIME Centre Wales (http://www.primecentre.wales). This included allowing core team members time to design, set up, and conduct the baseline and 3-month data collection. Financial support was provided by internal Cardiff Metropolitan University ‘Get Started’ and Cardiff University Division of Population funds to support transcription of the baseline qualitative data. PRIME Centre Wales, HealthWise Wales and the Centre for Trials Research are part of Health and Care Research Wales infrastructure (https://healthandcareresearchwales.org). Health and Care Research Wales is a networked organisation supported by Welsh Government. In August 2020, a Sêr Cymru III Tackling COVID-19 grant (https://gov.wales/ser-cymru, Project number WG 90) was awarded to cover our follow-up data collection, analysis and dissemination activities for the period between the 1st of August 2020 to 31st of March 2021. This work is supported by Health Data Research UK, which receives its funding from HDR UK Ltd (HDR-9006) funded by the UK Medical Research Council, Engineering and Physical Sciences Research Council, Economic and Social Research Council, Department of Health and Social Care (England), Chief Scientist Office of the Scottish Government Health and Social Care Directorates, Health and Social Care Research and Development Division (Welsh Government), Public Health Agency (Northern Ireland), British Heart Foundation (BHF) and the Wellcome Trust. The funders had no role in study design, data collection and analysis, decision to publish, or preparation of the manuscript.”

Additional Editor Comments (if provided):

Please carefully consider all comments in revising your manuscript.

Reviewers' comments:

Reviewer's Responses to Questions

**Comments to the Author**

1. Is the manuscript technically sound, and do the data support the conclusions?

Reviewer #1: Yes

Reviewer #2: Yes

2. Has the statistical analysis been performed appropriately and rigorously? 

Reviewer #1: N/A

Reviewer #2: N/A

3. Have the authors made all data underlying the findings in their manuscript fully available?

Reviewer #1: No

Reviewer #2: Yes

4. Is the manuscript presented in an intelligible fashion and written in standard English?

Reviewer #1: Yes

Reviewer #2: Yes

5. Review Comments to the Author

Reviewer #1: Overall, this manuscript was easy to read, with very basic content (basic descriptive statistics of a survey and planned follow-ups). For me, the lack of any relevant research question or insightful results was disappointing. I feel that a lot of this content could be put into the supplemental materials of a manuscript that reports novel findings. Currently, it feels like the manuscript is missing the results section. I feel the content presented could be used for the planned study “to identify the characteristics of people who have low engagement with infection-transmission prevention behaviour at different stages in the pandemic, with an emphasis on understanding the role of COVID-19 risk perception in determining behaviour.” Some of the content seems more relevant to a grant application than a research paper (e.g., “We presented early findings from the COPE study to the Welsh Government Technical Advisory Group on COVID-19 in April 2021. We will continue to engage with stakeholders nationally and internationally as the project progresses. Guided by our PPI members, we will disseminate public-facing summaries, infographics, and videos via the project social media feeds and website.”) I have added some specific comments that might improve the manuscript when the authors discover interesting findings for the scientific community.

“Use of physical barriers”, I was unsure what exactly this was referring to (e.g., quarantining, face mask, face shields, all of these).

Design section: I would break it up into shorter sentences.

Setting section: “within 28 days of a positive covid-19 test in the UK”. Perhaps say the first positive covid-19 test in the UK

It would be nice to clarify why you targeted under-represented demographics (motivations).

Not sure whether you write out what the following acronym means: SF12v1.

Some content of the ethics seems to be repeated. This level of detail seems unnecessary for a published piece.

In Table 2, ONS is not written out (Office of National Statistics).

Best of luck with the follow-ups.

Reviewer #2: Overall, the manuscript is well written, which I enjoy reading. I only have a few minor suggestions for the authors.

First, please provide (in a supplementary file) the name of the assessments used to measure different variables/topics listed in Table 1. For example, which scales are used to measure the attitudes towards COVID-19 vaccination, the caring responsibilities, work environment, etc. This might facilitate standardization of assessments, collaboration with other research groups, and comparison with other populations.

Second, according to Figure 2, explain what if the objective to compare COPE data with the population data for Wales and the UK. Also, in the discussion, you can provide some thoughts related the comparaison with population data for Wales and the UK.

Finally, in the discussion, you can provide some thoughts regarding PPI.

I wish the authors every success in their study.

6. PLOS authors have the option to publish the peer review history of their article (what does this mean?). If published, this will include your full peer review and any attached files.

Reviewer #1: No

Reviewer #2: **Yes: **François Routhier

---

## [Author Response · Author response to Decision Letter 0]

26 Sep 2021

We have removed all funding information from the manuscript as requested. We confirm that the following statement should be included in the Funding Statement section online:

“The initial stages of this research (March 2020 – August 2020) were supported by internal resources at Cardiff Metropolitan University (www.cardiffmet.ac.uk), Cardiff University (www.cardiff.ac.uk), HealthWise Wales (https://www.healthwisewales.gov.wales), and PRIME Centre Wales (http://www.primecentre.wales). This included allowing core team members time to design, set up, and conduct the baseline and 3-month data collection. Financial support was provided by internal Cardiff Metropolitan University ‘Get Started’ and Cardiff University Division of Population funds to support transcription of the baseline qualitative data. PRIME Centre Wales, HealthWise Wales and the Centre for Trials Research are part of Health and Care Research Wales infrastructure (https://healthandcareresearchwales.org). Health and Care Research Wales is a networked organisation supported by Welsh Government. In August 2020, a Sêr Cymru III Tackling COVID-19 grant (https://gov.wales/ser-cymru, Project number WG 90) was awarded to cover our follow-up data collection, analysis and dissemination activities for the period between the 1st of August 2020 to 31st of March 2021. This work is supported by Health Data Research UK, which receives its funding from HDR UK Ltd (HDR-9006) funded by the UK Medical Research Council, Engineering and Physical Sciences Research Council, Economic and Social Research Council, Department of Health and Social Care (England), Chief Scientist Office of the Scottish Government Health and Social Care Directorates, Health and Social Care Research and Development Division (Welsh Government), Public Health Agency (Northern Ireland), British Heart Foundation (BHF) and the Wellcome Trust. The funders had no role in study design, data collection and analysis, decision to publish, or preparation of the manuscript.”

Our response to the reviewers’ comments are as follows: 

• Data availability statement: Reviewers 1 and 2 have provided different responses. We have checked the information provided with the original submission and confirm that this is correct in that data is not publicly available. 

Reviewer #1: 

1. Overall, this manuscript was easy to read, with very basic content (basic descriptive statistics of a survey and planned follow-ups). For me, the lack of any relevant research question or insightful results was disappointing. I feel that a lot of this content could be put into the supplemental materials of a manuscript that reports novel findings. Currently, it feels like the manuscript is missing the results section. I feel the content presented could be used for the planned study “to identify the characteristics of people who have low engagement with infection-transmission prevention behaviour at different stages in the pandemic, with an emphasis on understanding the role of COVID-19 risk perception in determining behaviour.” 

Response: The purpose of this Cohort Profile manuscript is to provide detailed information on the COPE Cohort to provide essential information on the characteristics of the cohort and how these compare with the general population in Wales and the UK. This can then be referred to in future planned articles using data from this cohort to address specific research questions. This will provide greater transparency on our methods and the features of our cohort to facilitate replicability, collaboration, and comparison as we move forward with our study. The results of the analysis of determinants of infection transmission prevention behaviour in the cohort will be presented in full elsewhere. 

2. Some of the content seems more relevant to a grant application than a research paper (e.g., “We presented early findings from the COPE study to the Welsh Government Technical Advisory Group on COVID-19 in April 2021. We will continue to engage with stakeholders nationally and internationally as the project progresses. Guided by our PPI members, we will disseminate public-facing summaries, infographics, and videos via the project social media feeds and website.”)

Response: We have removed the section on collaborations and dissemination activities from the manuscript in response to this comment as on reflection this is probably not relevant in this context.

3. “Use of physical barriers”, I was unsure what exactly this was referring to (e.g., quarantining, face mask, face shields, all of these).

Response: We have now defined this term at first use in the introduction section to specify that we are referring to use of items such as such as face coverings, face shields, protective clothing, and disposable gloves. 

4. Design section: I would break it up into shorter sentences.

Response: We have broken the long sentence in this section into two shorter sentences. 

5. Setting section: “within 28 days of a positive covid-19 test in the UK”. Perhaps say the first positive covid-19 test in the UK

Response: We have re-phrased this sentence as follows to make the meaning clearer: “On the 13th of March 2020 when the COPE baseline survey was launched, within the UK there had been 480 confirmed cases and 16 reports of people having died within 28 days of having had a positive COVID-19 test”. 

6. It would be nice to clarify why you targeted under-represented demographics (motivations).

Response: Based on research in previous pandemics, we anticipated that key demographic variables such as age and gender would be important in understanding the impact of the pandemic on individuals and their responses to it. It became apparent as recruitment progressed that men and younger age groups were under-represented in our sample. We therefore decided to focus our limited advertising budget on reaching these groups. We have now clarified this in the manuscript. 

7. Not sure whether you write out what the following acronym means: SF12v1.

Response: This refers to the 12-item version of the Short-Form 12 Survey (Ware et al, 1996) and has now been defined at first use in the manuscript. 

8. Some content of the ethics seems to be repeated. This level of detail seems unnecessary for a published piece.

Response: We have reduced the level of detail provided on ethics and governance in the revised manuscript, while retaining essential information on approvals for the original study and subsequent amendments to enable data linkage and long-term follow-up. 

9. In Table 2, ONS is not written out (Office of National Statistics).

Response: We have amended this in the table. 

Reviewer #2: 

10. Overall, the manuscript is well written, which I enjoy reading. I only have a few minor suggestions for the authors. First, please provide (in a supplementary file) the name of the assessments used to measure different variables/topics listed in Table 1. For example, which scales are used to measure the attitudes towards COVID-19 vaccination, the caring responsibilities, work environment, etc. This might facilitate standardization of assessments, collaboration with other research groups, and comparison with other populations.

Response: We have added a data dictionary for the study as supporting information (S2_File), which is an Excel file detailing all the items included and citations for sources where applicable. 

11. Second, according to Figure 2, explain what if the objective to compare COPE data with the population data for Wales and the UK. 

Response: Table 2 includes a summary of Welsh and UK population data available from published sources to enable comparison of the COPE cohort with the general population and identification of under-represented demographic groups within the cohort. This has now been clarified in the manuscript text. 

12. Also, in the discussion, you can provide some thoughts related the comparison with population data for Wales and the UK.

Response: This is discussed in the ‘Limitations’ section of the discussion. We have reviewed this section and provided additional detail on how the characteristics of our sample need to be considered during analysis and interpretation of the data. 

13. Finally, in the discussion, you can provide some thoughts regarding PPI.

Response: We have added a paragraph to the ‘Future plans’ section of the discussion reflecting on the contribution our patient and public partners have made to the project so far and our plans for future involvement.

---

## [Editor Report · Decision Letter 1]

29 Sep 2021

Cohort profile: The UK COVID-19 Public Experiences (COPE) prospective longitudinal mixed-methods study of health and well-being during the SARSCoV2 coronavirus pandemic

PONE-D-21-22443R1

Dear Dr. Phillips,

We’re pleased to inform you that your manuscript has been judged scientifically suitable for publication and will be formally accepted for publication once it meets all outstanding technical requirements.

Kind regards,

Ismaeel Yunusa, PharmD, PhD

Academic Editor

PLOS ONE
---

## [Editor Report · Acceptance letter]

4 Oct 2021

PONE-D-21-22443R1 

Cohort profile: The UK COVID-19 Public Experiences (COPE) prospective longitudinal mixed-methods study of health and well-being during the SARSCoV2 coronavirus pandemic 

Dear Dr. Phillips:

I'm pleased to inform you that your manuscript has been deemed suitable for publication in PLOS ONE. Congratulations! Your manuscript is now with our production department. 

Kind regards, 

on behalf of

Dr. Ismaeel Yunusa 

Academic Editor

PLOS ONE